# Influence of Enzyme Additives on the Rheological Properties of Digester Slurry and on Biomethane Yield

**DOI:** 10.3390/bioengineering7020051

**Published:** 2020-06-04

**Authors:** Liane Müller, Nils Engler, Kay Rostalsky, Ulf Müller, Christian Krebs, Sandra Hinz

**Affiliations:** 1DBFZ Deutsches Biomasseforschungszentrum gemeinnützige GmbH, Biochemical Conversion Department, Torgauer Straße 116, 04347 Leipzig, Germany; liane.mueller@dbfz.de (L.M.); ulf.mueller@dbfz.de (U.M.); christian.krebs@dbfz.de (C.K.); 2Repowering Technik Ost GmbH, Reideburger Str. 43, 06116 Halle (Saale), Germany; k.rostalsky@repowering-technik-ost.de; 3DUPONT/ Genencor International BV, Archimedesweg 30, 2333 CN Leiden, The Netherlands; sandra.hinz@dupont.com

**Keywords:** anaerobic digestion, enzyme application, rheology of digestate

## Abstract

The use of enzyme additives in anaerobic digestion facilities has increased in recent years. According to the manufacturers, these additives should increase or accelerate the biogas yield and reduce the viscosity of the digester slurry. Such effects were confirmed under laboratory conditions. However, it has not yet been possible to quantify these effects in practice, partly because valid measurements on large-scale plants are expensive and challenging. In this research, a new enzyme product was tested under full-scale conditions. Two digesters were operated at identic process parameters—one digester was treated with an enzyme additive and a second digester was used as reference. A pipe viscometer was designed, constructed and calibrated and the rheological properties of the digester slurry were measured. Non-Newtonian flow behavior was modelled by using the Ostwald–de Baer law. Additionally, the specific biomethane yield of the feedstock was monitored to assess the influence of the enzyme additive on the substrate degradation efficiency. The viscosity measurements revealed a clear effect of the added enzyme product. The consistency factor K was significantly reduced after the enzyme application. There was no observable effect of enzyme application on the substrate degradation efficiency or specific biomethane yield.

## 1. Introduction

Anaerobic digestion (AD) is an established technology to gain energy from biogenic materials, agricultural residues and biowaste. By end of 2016, there were approximately 8700 biogas plants in operation in Germany [1], and the total number in Europe exceeds 18,200 [2].

The majority of biogas digesters currently operating in Europe are designed as continuously stirred tank reactors (CSTR). This reactor type requires more or less permanent homogenization and is therefore equipped with agitators. A broad variety of mixer types and mixing techniques have been developed and implemented, depending on the reactor size, feedstock and operation procedure. Nevertheless, mixing is still an issue and technical problems with mixers and stirrers came second in a survey of technical problems at biogas facilities in Germany [3,4].

Studies [5,6] have shown that the electricity required for agitation accounts for approximately 30–50% of the total electricity consumption of a typical biogas plant. This is equivalent to 5–9% of the electricity produced. Optimizing the efficiency of stirring and the minimizing energy demand of stirrers is therefore an important issue for the further optimization of the efficiency of biogas digesters.

Additionally, a correct determination of hydraulic retention time (HRT), an important process parameter, assumes that the digester volume is used at its full capacity. Incomplete homogenization due to improper mixing leads to dead zones inside the digester and hence to a reduced working volume and, as a consequence, to a reduced HRT. Therefore, proper stirring is fundamental for this type of reactor in order to achieve high biogas and methane yields as well as process stability [7].

A number of additives to optimize the anaerobic digestion process have been developed in recent years. Amongst them are enzyme mixtures explicitly designed to improve the biogas yield and affect the slurry rheology in a positive way. Producers claim that these additives can improve the methane yield and substrate degradation efficiency and lead to a reduced energy demand for mixing and pumping. Reliable data on the effectiveness of enzyme application in full-scale biogas digesters are however scarce, not least because measurements under practical conditions are very difficult.

Due to the non-Newtonian rheology of the digester slurry, a simple determination of the viscosity is not possible and more sophisticated approaches are necessary. In particular, the equations developed by Ostwald/de Waele and Herschel–Bulkley are commonly used to describe the rheological characteristics of shear-thinning fluids, such as sewage sludge or digester slurry [8]. Although the Ostwald–De Waele model is not meaningful from a physical point of view, (apparent viscosity goes towards zero for very high shear rates), it can provide a good agreement with practice for limited and defined ranges of validity. Moreover, the coefficients required for this model can be well determined experimentally. The Ostwald/de Waele approach is therefore widely used [9,10].

Research on the rheology of digester slurry has gained more attention in recent years. Many authors have worked on the characterization of AD digester slurry. The works of Baudez et al. [11,12] focused on sewage sludge and revealed that for different flow regimes (which means different shear rates), different models are suitable to describe the rheology. Other authors studied the influence of origin of digester slurry on rheological parameters. According to these findings, the rheology of digester slurry strongly depends on its origin and several process parameters, such as total solids content (TS), particle size and particle size distribution [13]. Furthermore, the type and configuration of the measurement device are of great importance because digester slurry is a non-Newtonian three-phase fluid containing liquid, gas bubbles and solid particles, partly in the form of longer fibres. Rotational viscometers in a classical cup-and-bob configuration have been used by some authors [11]. These devices turned out to be less suitable when the slurry contains larger particles or longer fibre structures. However, the removal of the interfering particles alters the sample in a way that is not permissible [14]. Agitator viscometers have been used by other authors facing more or less the same problems [15]. Misleading or inconsistent results are possible due to inappropriate measurement and experimental setup as well as interpretation of data [16]. There is still a lack of reliable, validated and practically applicable methods to determine the rheological properties of digester slurry.

The effect of enzyme supplementation on the AD process is controversially discussed. The role of enzymes in ruminal digestion processes is quite clear, and a significant fibre disintegration effect has been scientifically proven [17]. However, the process conditions—in particular the pH and microbial community—in the bovine digestive tract are very different from those in biogas digesters. Investigations into the stability of enzymes in AD facilities showed a very fast degradation and implied a neglectable effect [18]. An improved degradation and solubilization of lignocellulose material was observed; this, however, did not lead to an increase in the specific methane yield [19]. Other authors [20,21,22] have reported a significant increase in the methane yield of energy crops by enzyme application in lab digesters. Although the effect of enzyme application on the substrate degradation efficiency is still matter of discussion, plant operators frequently confirm positive overall effects, like the improved flow behaviur of the digester slurry, reduced floating layers and reduced sedimentation. However, they have been without scientific evidence until now.

This work aims at obtaining reliable measurement results on the influence of enzyme application on the biogas yield and viscosity of digester slurry under the conditions of AD plant operation in practice.

## 2. Materials and Methods

### 2.1. Anaerobic Digestion Tests

The experiments were carried out at a full-scale biogas research plant situated at Deutsches Biomasseforschungszentrum gemeinnützige GmbH (DBFZ) in Leipzig. The research biogas plant comprises two continuously stirred tank reactors (CSTR) with a gross volume of 208 m³ each. They are equipped with a common feeding system that allows adding solid feedstock via a solid feeder, as well as liquid feedstock from a separate storage tank of 174 m³. The standard operation mode is at a mesophilic temperature (39 °C). The plant is equipped with extensive measuring technology for all the relevant process parameters, such as feedstock mass flow, process temperature, gas flow and gas composition. A schematic plant design is shown in Figure 1.

In order to assess the influence of enzyme application, both digesters of the research plant were operated in parallel mode. The operating volume was 189 m³ for each digester during the experiment. The feedstock was a mixture of sorghum silage and cattle manure (40%/60% mass, based on fresh matter). The cattle manure originated from a cattle farm located near Leipzig. Ten batches of cattle manure were delivered during the trial period. Due to unforeseen shortages, it was necessary to change the supplier for the sorghum silage during the trial period. The silages with different origins are identified below as Sorghum silage I and Sorghum silage II. Four batches of Sorghum silage I and five batches of Sorghum silage II were used for the experiment.

Digester 3.1 received the feedstock mix described above and additionally the enzyme additive described in Section 2.2. Digester 3.2 was used as a reference and therefore received the same substrate mixture without the enzyme additive. The organic loading rate was maintained at 5.0 kg_VS_/(m³·d)^−1^ for both digesters, resulting in an HRT of 25 days on average. After a pre-period of 35 days (1.5× HRT), the enzyme application started in digester 3.1 and was maintained over 97 days (approx. 4× HRT). Subsequently, a phase-out period of 28 d (1× HRT) followed. All the described process parameters were monitored over the entire trial period. The feedstock mass flow, process temperature, gas flow and gas composition were controlled online. The process parameters total solids (TS), volatile solids (VS), pH, volatile fatty acids (VFA), alkalinity and ammonia were controlled in order to evaluate the process stability.

### 2.2. Enzyme Product

The enzyme product used in this study is a formulation developed within the joint research project *DEMETER—*demonstrating more efficient enzyme production to increase biogas yields [23]. This research project aimed to develop an efficient enzyme production process and demonstrate effects of enzyme application in a full-scale operation. The enzyme product contains cellulases and xylanases, which all originate from the strain Myceliophthora thermophila C1. The dosage carried out according to the manufacturer’s specifications was 0.45 g per kg_VS_ of substrate.

### 2.3. Analytic Methods

The feedstock and digester slurry were regularly analyzed. The total solids (TS) and volatile solids (VS) were analyzed according to the European standard DIN EN 15934 and DIN EN 15935. The Weender method was used to determine the crude fat (XL), crude protein (XP) and crude fibre (XF). The process stability parameters were measured two times per week. The VFA and alkalinity were determined through titration using an automatic titration system, Mettler Toledo Rondo 60/T90 (Mettler-Toledo gmbH Gießen, Germany). The ammonia nitrogen was analyzed in the liquid phase of the digester slurry by the photometric method (Spectrophotometer DR 3900, Hach Lange GmbH, Germany). All the methods mentioned above are described in detail in a collection of measurement methods for biogas [24]. The specific methane potential was measured in batch tests in lab scale (1 L) using the Automatic Methane Potential Test System (AMPTS) (Bioprocess control, Sweden). Tests were carried out at a mesophilic temperature (38 °C). The test protocol followed the German guideline VDI 4630, which implements the recommendations of a European expert commission [25]. The feedstock analyses are shown in Table 1.

### 2.4. Pipe Viscometer

A pipe viscometer (see Figure 2) was developed over the course of the project and used to assess the rheological properties of the digester slurry. The slurry is drawn from one of the digesters and stored in a tempered reservoir tank (V = 1000 L). A progressive cavity pump with a controlled speed is used to convey the slurry into the viscometer, which consists of an inlet zone to obtain the laminar flow and a metering section equipped with two pressure sensors mounted at a distance of 8 m, followed by an outflow zone. Three different pipe diameters (65, 80 and 100 mm) are available in order to adapt the flow behavior to different viscosities. The volume rate is measured by a magneto-inductive flowmeter (PROMAG 55; Endress + Hauser; Switzerland), positioned in the inlet zone. After passing the outlet zone, the medium is transported back to the reservoir tank. A number of ball valves allow the purging of the entire pipeline with fresh material to remove any residues from previous measurements.

For the validation and calibration of the viscometer, a commercial capillary viscometer type Rheotest 2 (Rheotest Medingen GmbH, Germany) is used as a reference. Non-Newtonian flow behavior is assumed for the digester slurry, and therefore validation runs are conducted with xanthan solution, which shows a similar shear-thinning and thixotropic flow characteristics [26]. The xanthan solution was used at three different concentrations (200, 250 and 287 g·L^−1^), representing three different levels of apparent viscosity.

Measurements with the pipe viscometer followed a fixed procedure. After filling the reservoir tank with fresh slurry from the digester, samples for TS and VS analyses were taken. The temperature was kept constant (39 ± 2 °C) over the entire measuring procedure by an external heating jacket, and the homogeneity was maintained by an internal mixer. Before measurement, the pipeline system was flushed with fresh digester slurry and the losses were replenished. The measurement procedure included 5 min pre-shearing at the lowest shear rate, followed by a stepwise increase in the shear rates. Each shear rate was maintained for 5 min, and the pump drive speed, volume rate, pressure and pressure drop over the metering section were recorded every second. The procedure was repeated in triplicate.

For each shear rate, the apparent viscosity was calculated by Equation (1):(1)$$\eta =\frac{\Pi \cdot \Delta_{P}\cdot D^{4}}{128\cdot Q_{v}\cdot L},$$
where:*η* apparent viscosity in Pas;*Δp* pressure drop over *L* in Pa;*D* diameter of metering section in m;*Q_v_* volume rate in m³·s^−1^;*L* distance between pressure sensors in m.

With the apparent viscosity, the Reynolds number was calculated according to Equation (2) to ensure that the flow regime was laminar.
(2)$$Re=\frac{uD\rho }{\eta },$$
where:*Re* Reynolds number;*η* apparent viscosity in Pas;*ρ* density in kg·m^−3^;*D* diameter of metering section in m;*u* flow velocity in m·s^−1^.

The apparent viscosity was displayed over the shear rate in a flow diagram. The Ostwald–de Baer law as shown in Equation (3) is suitable to describe the shear-thinning behavior [9] and was used to determine the rheological parameters.
(3)$$\eta =K\cdot \;{\dot{\gamma }}^{\left(n-1\right)},$$
where:*η* apparent viscosity in Pa·s;$\dot{\gamma }$shear rate in s^−1^;*K* consistency factor in Pa·s^n^;*n* flow index.

The parameters *K* and *n* were calculated by plotting the logarithmized values for the apparent viscosity log*η* against the logarithmized corresponding shear rate $\log\dot{\gamma }$ and performing linear regression. The procedure for measuring and data processing was the same for the control measurement with xanthan as well as for the measurements with digester slurry during the enzyme application.

Unfortunately, the pipe viscometer was not available during the pre-trial phase, and therefore no data for this period are available.

### 2.5. Substrate Degradation Efficiency

The substrate degradation efficiency (SDE) was used as an assessment parameter to evaluate the effects of enzyme application on the biomethane yield. The biomethane potential (BMP) of a specific substrate defines the maximum amount of methane that can potentially be produced during AD. The BMP can be approximated based on the chemical substrate composition, stoichiometric calculations or discontinuous digestion (anaerobic batch or BMP) tests. Due to diverse metabolic pathways during biochemical conversion, a certain amount of substrate is also utilized for microbial growth or maintenance, which consequently lowers the BMP. Furthermore, substrate pre-treatment or disintegration can change the BMP of the investigated substrate [27]. The methane yield describes the utilized fraction of the methane potential under practical conditions during full-scale (or laboratory) continuous operated anaerobic digestion. Thus, the yield depends on numerous impact factors, such as retention time, organic loading rate, inhibitory effects or nutrient deficiency. The SDE can be calculated as the quotient of the methane yield over the BMP:(4)$$SDE=\frac{\;methane\;yield}{methane\;potential}.$$

By definition, the specific methane yield has to be lower or equal to the BMP. Thus, the degradation efficiency SDE should be in a range between 0 and 1. The specific BMP was calculated in accordance to a method published by Weißbach [28]. This method allows to us calculate the BMP of silages based on the crude fibre content. The method is described in [24]. The methane yield was measured during the trials at the DBFZ research plant described above.

## 3. Results and Discussion

### 3.1. Specific Biogas Yield and Substrate Degradation Efficiency

The experiment started with a preparation period of 35 days in which both digesters were operated identically in order to establish uniform process conditions. Steady state was defined by a uniform and constant specific gas production over at least 1 HRT and was proven by Neumann trend test statistics for both of the digesters separately [29]. The application of C1 enzymes started at day 120 in digester 3.1, whilst digester 3.2 was used as reference—i.e., with the same process conditions (same substrate and organic loading Rate (OLR)) but without enzyme application. The enzyme application was maintained over 97 days (approx. 4 hydraulic retention times (HRT)). Subsequently, a phase-out period of 20 d (0.8 HRT) followed. The process parameters as described above were monitored over the entire trial period and are shown in Table 2.

Unfortunately, the sorghum silage quality changed between the used batches. Table 3 shows the differences between the two batches of sorghum silage used during the experiment. The TS and VS contents as well as the specific methane potential differ significantly. As a result, the specific biomethane yield of the mix (sorghum silage + cow manure) varied during the experiment. The biomethane potential and biomethane yields were calculated based on the input data. For a comparison of the viscosity, time periods were selected with the same feeding quality—i.e., for the same batch of sorghum silage.

The increase in specific methane yield during the phase of enzyme addition occurred in both digesters to the same extent. It can therefore clearly be attributed to the change in feedstock quality rather than to enzyme application. To enable a comparison of both digesters independent of the specific methane yield of the input, the relative deviation—i.e., the difference between both digesters in relation to the reference, digester 3.2—was calculated. The result is shown in Figure 3. As can be seen, the difference between digester 3.1 with enzyme application and digester 3.2 (reference) is within the range of ±10% over the entire experiment. The relative deviation remains in the same range during the phases with and without enzyme application in digester 3.1. There is no statistically significant trend, as confirmed by a Neumann trend test at a level of significance of 95%.

Both of the digesters showed no indications of instabilities, all process parameters remained in a normal range and no differences between the digester with enzyme application and the reference digester were observed throughout the entire experiment (see also Table 2).

### 3.2. Calibration of Pipe Viscometer

Xanthan was used in three concentrations—200, 250 and 287 g·L^−1^—to validate and calibrate the pipe viscometer. The flow curves of all three concentrations were measured by the reference capillary viscometer (CV) in duplicate and the pipe viscometer in triplicate. The results for the consistency factor K and flow index n are shown in Figure 4 and Figure 5.

As expected, the consistency factor K depends on the Xanthan concentration. Furthermore, the calculated value of K increases with an increased pipe diameter. This effect can be explained by the different flow profiles. The Reynolds number was in the range of 1600 (2.87% Xanthan) to 2800 (2% Xanthan) for the 65 mm pipe diameter. For the 100 mm pipe diameter, the Reynolds number ranged between 400 (2.87% Xanthan) and 680 (2% Xanthan). Hence, a non-laminar flow was assumed for the 65 mm pipe diameter, and the 100 mm pipe was used for further measurements. A comparison of the results gained by both viscometers is shown in Table 4. The difference between the capillary viscometer, used as a reference, and the pipe viscometer with a diameter of 100 mm is less at higher absolute values of K—i.e., at a higher apparent viscosity.

Measurements with the pipe viscometer showed a high reproducibility. The viscometer is therefore suitable to detect changes in the apparent viscosity of digester slurries with or without enzyme application. As the apparent viscosity of the digester slurry is expected to be in the higher range (K > 30), all the following measurements were carried out with the 100 mm pipe viscometer.

### 3.3. Rheological Characteristics of Digester Slurry

Flow curves of the digester slurry from both digesters (3.1/enzyme application, 3.2/reference) were measured three times after the start of the enzyme application. A typical flow diagram showing the apparent viscosity vs. the shear rate is shown in Figure 6. As can be seen from the regression parameters, the approach shown in Equation (3) delivers a good accordance with the data.

The results of measurements are shown in Table 5. The consistency factor K is far lower for the slurry from digester 3.1. This means that the general flow properties have been improved and the apparent viscosity of the digester slurry is far lower with enzyme application than without. The difference in the flow index n is low but reproducible throughout all the measurements. It is slightly higher for the slurry from the digester with enzyme application. This result implies that the non-Newtonian behavior—i.e., the increased shear rate depending shear thinning—is more apparent for the digester with the enzyme additive.

Additional measurements were conducted at day 122, which is 25 days (approx. 1 HRT) after the enzyme application had stopped. The parameters K and n (shown in last row of Table 5) had not increased after the termination of the enzyme application and remained at the level they were before. It is expected that this effect will subside over time. The apparent viscosity versus the shear rate is shown for the digester slurry with and without enzyme addition in Figure 7. The effect of the enzyme addition on the apparent viscosity is higher at lower shear rates. This effect almost disappears at higher shear rates.

The economic effects of reduced viscosity are mainly due to a lower energy demand for stirring processes. Studies [13] have shown that there is a strong correlation between the viscosity and energy consumption in biogas fermenters. A deep discussion of economic aspects would go beyond the scope of this manuscript. Energy requirement for mixing makes up a considerable part of the total energy requirement of biogas plants [30]. Hence, a positive economic effect of the enzyme application can be assumed.

## 4. Conclusions

The validation tests showed that a pipe viscometer is suitable to measure the rheological parameters of digester slurry under practical conditions. Validation and calibration with a non-Newtonian model fluid showed good accordance with the reference method (capillary viscometer) for a laminar flow regime at a 100 mm pipe diameter. Non-Newtonian flow behavior can be approximated by the Ostwald–de Baer equation with good accordance to the data. Measurements with digester slurry from a full-scale biogas research plant have confirmed the reproducibility of the results.

The experiment revealed the clear effects of enzyme application in full-scale operation. The enzyme product influenced the rheological parameters of the digester slurry. Slurry from the digester with enzyme application showed a significantly decreased consistency factor K compared to the reference digester. This indicates a generally decreased apparent viscosity. The flow index n is slightly higher for the slurry from the digester with enzyme application. This means that the dependence of the shear thinning on the shear rate was also influenced by the enzyme product. Thep possible economic effects are a reduced energy demand for mixers and pumps.

There was no observable effect of enzyme application on the substrate degradation efficiency or specific methane yield. Both the digesters with and without enzyme application showed nearly the same specific methane yields over the entire experimental period. Changes in the substrate degradation efficiency can be attributed to changes in the feedstock quality during the trial. The observed changes in the rheological parameters are apparently not associated with an improved methane yield.

## Figures and Tables

**Figure 1 bioengineering-07-00051-f001:**
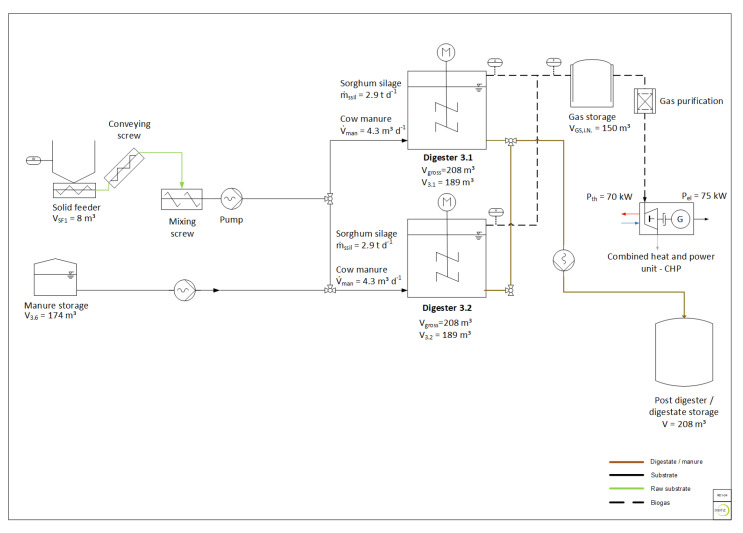
Scheme of the research biogas plant at Deutsches Biomasseforschungszentrum (DBFZ).

**Figure 2 bioengineering-07-00051-f002:**
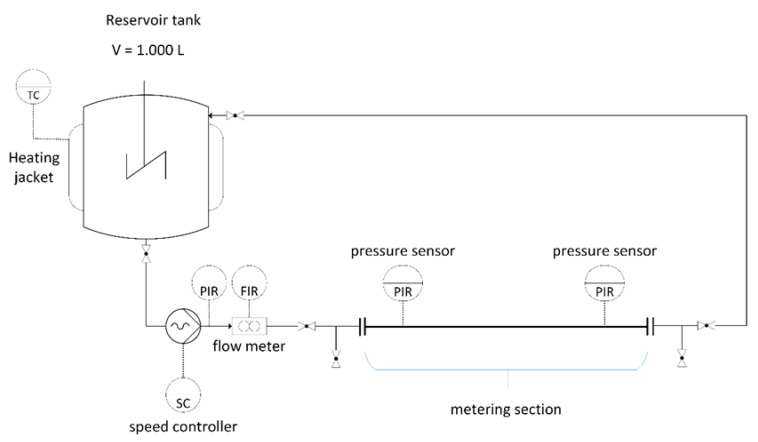
Scheme of the pipe viscometer.

**Figure 3 bioengineering-07-00051-f003:**
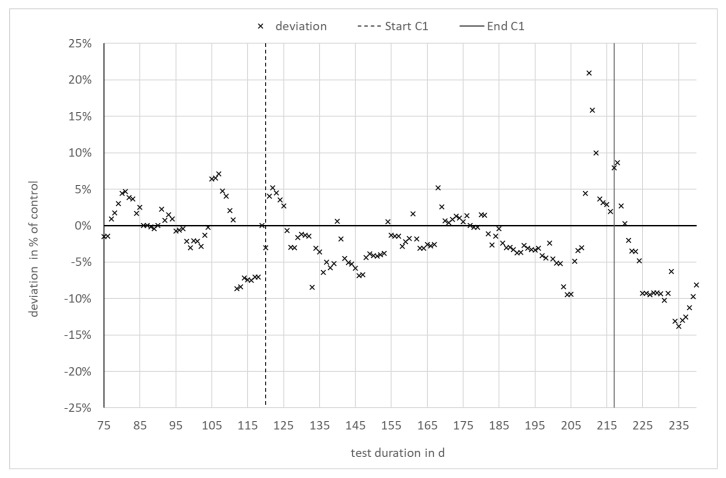
Deviation in specific methane yield in % of the reference digester 3.2.

**Figure 4 bioengineering-07-00051-f004:**
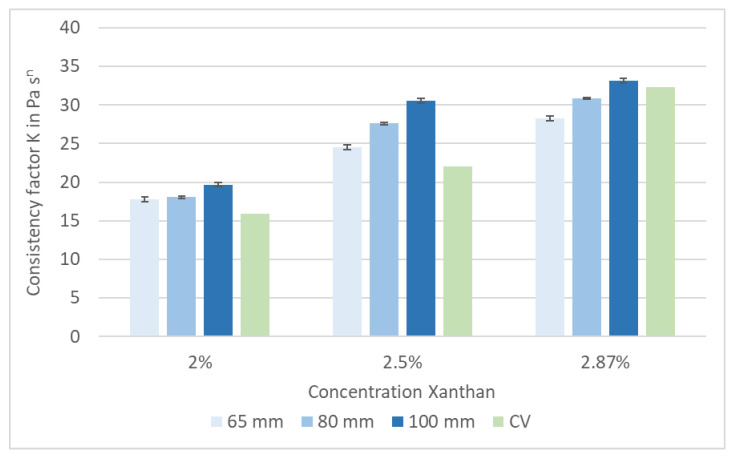
Xanthan solution: consistency factor K, measured with a pipe viscometer with different diameters and with a capillary viscometer (CV).

**Figure 5 bioengineering-07-00051-f005:**
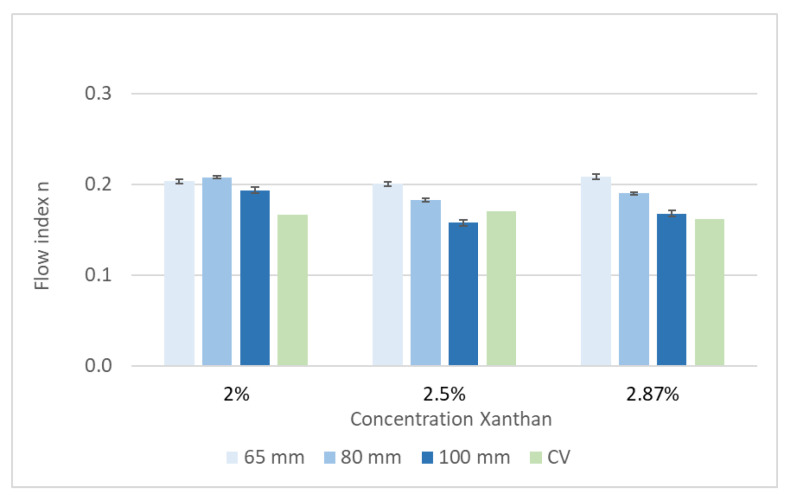
Xanthan solution: flow index n, measured with a pipe viscometer with different diameters and with a capillary viscometer (CV).

**Figure 6 bioengineering-07-00051-f006:**
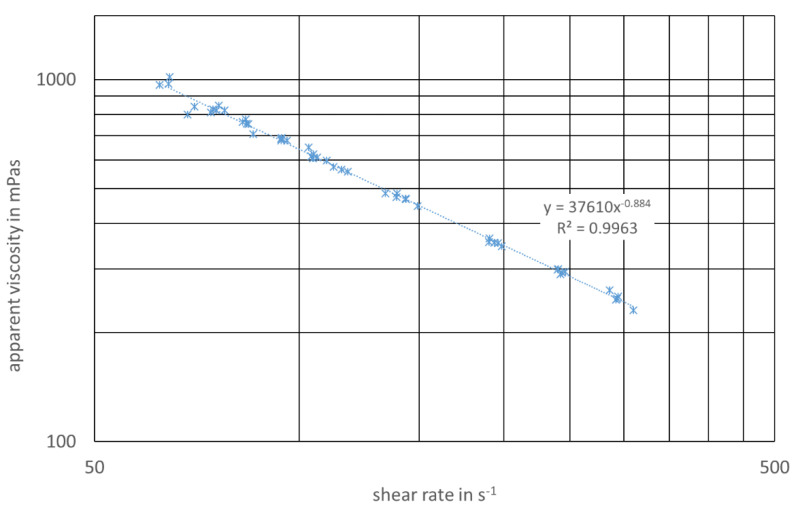
Flow diagram, digester 3.1/with enzyme application and regression parameters.

**Figure 7 bioengineering-07-00051-f007:**
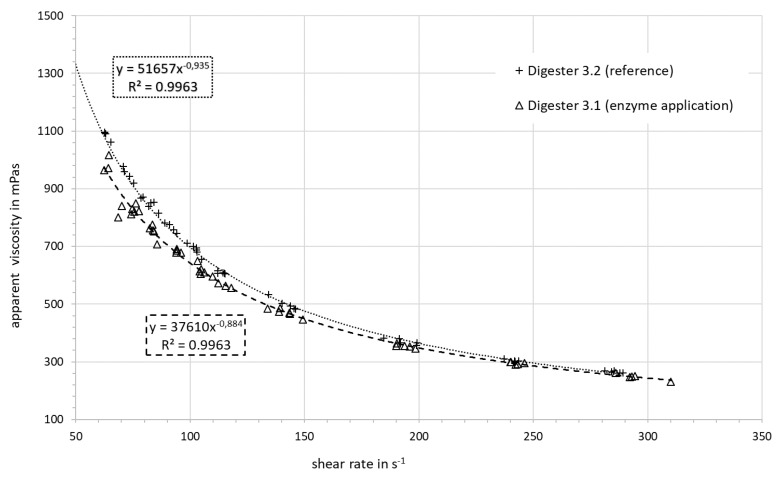
Apparent viscosity vs. the shear rate: comparison of digester slurry with and without enzyme addition.

**Table 1 bioengineering-07-00051-t001:** Feedstock analyses of the substrates used.

Substrate	TS	VS	Crude Fat	Crude Protein	Crude Fibre	CH_4_- Potential
	**%**	**%TS**	**g·kg^−1^ TS**	**g·kg^−1^ TS**	**g·kg^−1^ TS**	**L·kg^−1^ TS**
**Sorghum silage I (n = 4)**	28 ± 3	95 ± 1	20 ± 3	71 ± 7	436 ± 24	378 ± 8 (n = 3)
**Sorghum silage II (n = 5)**	1 ± 2	92 ± 1	27 ± 4	104 ± 10	412 ± 26	350 ± 7 (n = 3)
**Cattle manure (n = 10)**	9 ± 1	76 ± 1	37 ± 8	160 ± 18	259 ± 20	241 ± 9 (n = 3)

**Table 2 bioengineering-07-00051-t002:** Process parameters in digester 3.1 (enzyme addition) and 3.2 (reference) during different experimental phases.

Digester	pH	VFA g·L^−1^	NH_4_-N mg·L^−1^	TS %	VS % TS	Specific CH_4_-Yield m³·t^−1^
**Digester 3.2 (reference) (during pre-phase)**	7.6 ± 0.1 (n = 12)	2.0 ± 0.1 (n = 13)	2.0 ± 0.1 (n = 7)	10.4 ± 0.3 (n = 8)	79.5 ± 0.9 (n = 8)	226 ± 13 (n = 44)
**Digester 3.2 (reference) (during enzyme addition in digester 3.1)**	7.6 ± 0.1 (n = 18)	1.7 ± 0.1 (n = 19)	1.7 ± 0.1 (n = 14)	8.6 ± 0.4 (n = 12)	77.2 ± 1.1 (n = 12)	317 ± 24 (n = 68)
**Digester 3.2 (reference) (during post phase)**	7.6 ± 0.2 (n = 7)	1.7 ± 0.0 (n = 7)	1.7 ± 0.1 (n = 4)	8.7 ± 0.1 (n = 4)	75.7 ± 4 (n = 4)	337 ±17 (n = 20)
**Digester 3.1 (pre-phase)**	7.6 ± 0.1 (n = 13)	2.0 ± 0.1 (n = 13)	2.0 ± 0.1 (n = 7)	10.2 ± 0.3 (n = 8)	79.5 ± 0.7 (n = 8)	224 ± 15 (n = 44)
**Digester 3.1 (during enzyme addition)**	7.7 ± 0.1 (n = 19)	1.7 ± 0.1 (n = 19)	1.6 ± 0.1 (n = 14)	8.5 ± 0.4 (n = 12)	76.6 ± 1.2 (n = 11)	312 ± 33 (n = 68)
**Digester 3.1 (post-phase)**	7.6 ± 0.1 (n = 7)	1.7 ± 0.1 (n = 7)	1.7 ± 0.1 (n = 4)	8.5 ± 0.1 (n = 4)	76.4 ± 0.6 (n = 4)	318 ± 16 (n = 20

**Table 3 bioengineering-07-00051-t003:** Specific methane yields and substrate degradation efficiency.

Digester	Specific CH_4_-Yield of Mix (Sorghum Silage + Cow Manure) m³·t^−1^ VS	SDE (Equation (4)) %
**Digester 3.2 (reference)** **(during pre-phase)**	226	67
**Digester 3.2 (reference)** **(during enzyme addition in digester 3.1)**	322	94
**Digester 3.2 (reference)** **(during post phase)**	331	96
**Digester 3.1** **(pre-phase)**	231	68
**Digester 3.1** **(during enzyme addition)**	323	94
**Digester 3.1** **(post-phase)**	317	93

**Table 4 bioengineering-07-00051-t004:** Validation runs for the pipe viscometer (results for diameter 100 mm in triplicate).

	Pipe Viscometer D = 100 mm		Capillary Viscometer	
Xanthan Concentration	Consistency Factor K	Flow Index n	Consistency Factor K	Flow Index n
**2%**	20 ± 0,2	0.19 ± 0,003	16	0.17
**2.5%**	30 ± 0,3	0.16 ± 0,003	22	0.17
**2.87%**	33 ± 0,2	0.17 ± 0,002	32	0.16

**Table 5 bioengineering-07-00051-t005:** Consistency factors and flow indices of the digestate from the treated and untreated digester at different times.

Time	Digester 3.1/with Enzyme Application	Digester 3.2/Reference
from Starting Enzyme Application	Consistency Factor K	Flow Index n	Consistency Factor K	Flow Index n
86 d	38	0.12	52	0.06
90 d	38	0.12	51	0.07
93 d	36	0.12	53	0.07
from termination of enzyme application				
25 d	37	0.12

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
