# Peer review of "Influence of Enzyme Additives on the Rheological Properties of Digester Slurry and on Biomethane Yield"

_bioengineering, 2020, doi:10.3390/bioengineering7020051_

Round 1

Reviewer 1 Report

This paper presents valuable and original data on enzyme addition in full scale anaerobic digestion and its impact on digestate rheology and methane production. However, discussion should be significantly improved, in particular I am not convinced by the validation of the pipe viscometer. In addition, some information is lacking in material and methods.

I have a concern with the use of a single batch of cattle manure during the whole experimental period (160 days). It is known that methane potential of old manure is lower than methane potential of fresh manure. Was it taken into account? Table 1 shows 3 measures of methane potential, were they carried out at the same time?

The composition of the main enzymes (cellulases, hemicellulases, pectinase, laccases?) should be given in material and methods.

The dose of enzyme addition should be given.

As a non-German reader, I could not find information on BMP calculation. This information must be added in material and methods.

In general, authors should avoid non English references [1,3,5,22].

Did all the xanthan solutions led to laminar flow in all the pipes studied? This should be discussed in relation with the validity of the measurements.

In addition, it is not clear if the measures given by 100 mm pipe viscometer are valid. It seems that they are not, in particular K, please discuss.

There is no consistency between figure 5 (n=0.05) and table 3 (n=0.12) for treated digestate.

The authors should present results about apparent viscosity in the digester and quantify its reduction due to enzyme addition? What are the consequences on digester mixing cost and on digestate transportation cost? If possible, these costs should be compared to the enzyme costs.

Tables 4 and 5 may be merged.

There is some contradiction between lines 261-262 “the shear thinning effect is more apparent…with enzymes” and lines 316-317 ‘the shear thinning is not influenced by the enzyme product”

Line 116. Do not use capital letter for ammonia

Line 152. What is volume rate? The volume flow rate?

Line 179 im should be in

Line 182 use capital letter for Reynolds

Author Response

Dear Reviewer,

thank you very much for your review and the valuable remarks. The manuscript has undergone comprehensive revisions and the authors have made every effort to address all the points raised. Please find our responses in the attached document.

Reviewer 2 Report

The subject line of this work is very interesting; however it has been presented rather in the confusing manner.

Major comments

  1. The novelty is not clearly stated, neither in the "Abstract" nor at the end of "Introduction". Thus, the contribution to this paper is not being recognized well. 
  2. Page 3: Section 2. Materials and Method: What’s the technical size of the digesters? Mention the digester active capacity?
  3. Page 3: Section 2. Materials and Method: If this the introduction to the following sections 2.1 and so on, please give a clear picture of a story to summarise it. Actually its confusing this paragraph. I suggest the authors to move this paragraph to the respective sections (2.1-2.4). Also introduce one more sub-section for the analytic methods.
  4. Fig. 1: what’s the size of the post digestate or the storage tank? Also, the rate of recirculation and its frequency?
  5. The order of the sub-titles in the results section are not in accordance with the materials and method section. Probably 3.1 and 3.2 can be moved after 3.3.
  6. If I understand correctly, digester 3.1 received the feedstock mix and additionally the enzyme additive. Whereas, digester 3.2 act as a reference digester, which received no enzyme additive. If this statement is correct, why table 4 and 5 contains different explanation of the digesters. Why reference digester has the enzyme addition phase? Section 3.3 has to be rewritten. The phrases are confusing.
  7. In "Figure 2", the actual range of "deviation in % of control" lies between -15 % to +25% approximately. Why have the authors plotted the curve between -50% to +50%?
  8. How were the consistency factor (K) and the flow index (n) calculated as shown in figure 3 and 4, using equation 1, 2 and 3?; similarly is the case for Table 3.
  9. What is the concentration of the enzyme added, as its concentration is expected to affect K and n, following what is mentioned in table 2? Similarly, if it does, how does it affect the rheology?
  10. Again conclusion section is confusing. What are the major conclusions. Can you please restate this part based on your objective of the study. If both digesters, treated and control, showed nearly identic specific biomethane yields over the entire experimental period, what is the major break through of this study (economically, environmentally ?). More information should be included in the manuscript to support this argument.

Minor corrections

  1. Some places English writing is vague and unsystematic, some of the sentences formed are difficult to understand. The authors may consider to rewrite the article with clear explanations. 
  2. Sentence formation in the paper is grammatically incorrect, for example, line (15-18) "Enzyme additives, explicitly designed to affect slurry rheology in a positive way and to improve substrate degradation, have entered the market in the recent years. Reliable data on effectiveness of enzyme application in full-scale biogas digesters are however scarce, not at least because measurements under practical conditions are very difficult" are complex statements, therefore difficult to understand what authors want to convey. Likewise, the complete paper is written in a similar pattern. Therefore, a revision is required in language. 
  3. The paper is not cited in a proper manner. Hence, this further increases the complexity to the reader. As well as the citations are often mentioned after the sentence ends or after a full stop. Example line 50 "operation. [7]". This is found in the paper elsewhere too. 
  4. For line 41, the range is not correctly typed "approximately 30...50 %", it is the same for line 42. 
  5. The references are not stated in a proper way. 
  6. "Error! Reference source not found" is identified at many places, for example, line (283, 287,305, etc.). This needs to be corrected. 
  7. The last thing to mention is there are minor spelling mistakes at multiple places                                                        - Eg. Figure 3 and 4, concentration is written as 2,5%, instead of 2.5%                                                                    - Xanthan as Xanthane in Table 2 and likewise.

Author Response

Dear Reviewer,

Thank you very much for your review and the valuable remarks. The manuscript has been revised accordingly and the authors have made every effort to address the Points raised. Please find our answers to the specific comments in the attached document.

Round 2

Reviewer 1 Report

Most of the comments have been adressed. However, few issues remain:

line  322 "The flow index n was not affected" I do not think that one can affirm this comparing 0.12 versus 0.06/0.07. I rather agree with the statement lines 301-302 . It is slightly higher for the slurry from digester with enzyme application." 

To actually show that the apparent viscosity decreased after enzyme addition I suggest to add a table or a figure showing apparent viscosity with and without enzyme addition at different shear rates.

line 325"Scenario calculations in the framework of the DEMETER research project [30] showed that possible  energy savings can more than compensate for enzyme costs" I could not find the information on the projetc website, please provide a reference that can be checked, remove the sentence or provide arguements and calculations.

Author Response

Dear Reviewer,

thank you for the helpful remarks. Please find our comments and answers in the attached file.

Reviewer 2 Report

The authors carried out required corrections.

Author Response

Dear Reviewer,

We would like to thank you for the feedpack and are pleased that all the criticisms raised have been resolved to your satisfaction.